

# Reconstruction of wind and surge of the 1906 storm tide at the German North Sea Coast

Elke M.I. Meyer[1], Ralf Weisse[1], Iris Grabemann[1], Birger Tinz[2], Robert Scholz[2,3]

[1]Helmholtz-Zentrum Hereon, Institute of Coastal Systems, Geesthacht, 21502, Germany
[2]Deutscher Wetterdienst, Hamburg, 20359, Germany
[3]Deutscher Wetterdienst, Leipzig, 04288, Germany

*Correspondence to*: Elke M.I. Meyer (elke.meyer@hereon.de)

**Abstract**

Storm tides represent a major threat to the low-lying German North Sea coast. Knowledge of extremes is essential for the design of reliable and robust coastal defences. A storm tide that occurred on 12-13 March 1906 along the German Bight coastline still represents one of the strongest events on record. For this event, detailed knowledge of atmospheric and

hydrodynamic conditions is still lacking. To assess the potential impact of such an event on today's coastline, century-long atmospheric reanalysis data together with a manual synoptic reconstruction based on archived weather data were used to drive a tide-surge model and to simulate water levels during the event. Sensitivity experiments were performed to estimate potential amplification of water levels that could have been caused by different time lags between the storm and the astronomical tide. Comparison between the model results and the limited available observational data indicated, that the

water levels could be reasonably reconstructed using wind fields from the manual synoptic approach and some of the reanalysis ensemble members. The amplification potential was found to be low because the storm occurred during spring tide and shifts in the phase of the astronomic tide yielded only small changes in total water levels. To summarize, if pressure data are available at relevant locations, historical storm surges can be simulated with reanalysis products and also with a manual synoptic reconstruction.

**1 Introduction**

The German North Sea coast is characterized by mudflats with islands and lowlands and is particularly vulnerable to extreme storm tides. During storm surges, the offshore islands are both vulnerable and serve as barrier for the mainland behind (McBride et al. 2013). An almost continuous line of dykes with heights ranging between approx. 5-10 m protects the low-lying hinterland from potential storm tide impacts.

Knowledge on extreme storm tides such as their height, duration, or associated wind and wave fields, is therefore needed for the design and dimensioning of protective measures (e.g. Generalplan Küstenschutz (Master plan coastal protection), NLWKN, 2007), the optimization of operational (e.g. forecast) procedures (e.g. Streicher et al. 2015) or for risk assessments (e.g. Schaper et al. 2019).

During the night of 12-13 March 1906, a severe storm tide hit major parts of the German North Sea coastline and the coast

of Belgium, the Netherlands, and Denmark (van Bebber, 1906, Figure 1). It was caused by a coincidence of strong north-westerly winds that pushed waters towards the coast and high astronomical (spring) tides that in combination caused extreme





water levels. While more than hundred years have passed since then, the event is still responsible for some of the highest
measured water levels at some tide-gauges along the Lower Saxony coast (e.g. Emden, Table 1, Figure 1).

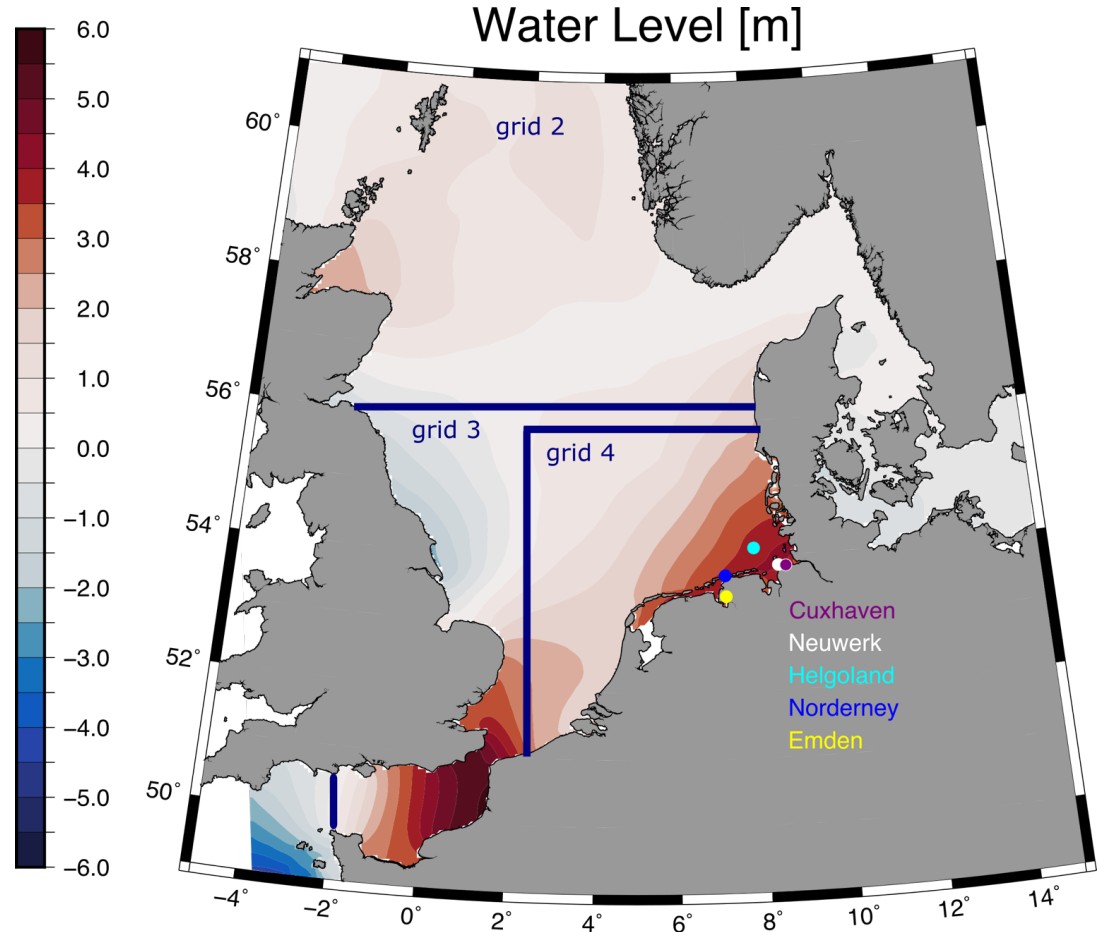

5   Figure 1: North Sea model regions of grid 2, 3 and 4 and locations of the signal stations and gauges. The colours present the
spatial distribution of simulated water levels at 13 March 1906, 0 UTC. The atmospheric forcing is taken from 20CRv3-
ensemble member 34.

| Emden | | Norderney | | Wilhelmshaven | | Cuxhaven | |
|---|---|---|---|---|---|---|---|
| Lon: 7.2° | Lat: 53.3° | Lon: 7.1° | Lat: 53.7° | Lon: 8.1° | Lat: 53.5° | Lon: 8.7° | Lat: 53.9° |
| HW [m NAP] | Date | HW [m NAP] | Date | HW [m NAP] | Date | HW [m NAP] | Datum |
| 5.18 | 1906-03-13 | 4.09 | 1962-02-16 | 5.18 | 1962-02-16 | 5.1 | 1976-01-03 |
| 5.17 | 2006-11-01 | 4.07 | 2013-12-06 | 5.08 | 2013-12-06 | : | : |
| 5.12 | 1944-02-04 | 3.95 | 1906-03-13 | 5.06 | 1906-03-13 | 4.36 | 1906-03-13 |

Table1: Observed high water (HW) levels above Normaal Amsterdams Peil (NAP) for selected tide gauges in the German
Bight (Deutsches Gewässerkundliches Jahrbuch, 2013). The first station is located in the Ems estuary, Norderney is an East-



Frisian Island, the gauge of Wilhelmshaven is situated in the Jade Bay, and the Cuxhaven gauge in the Elbe estuary. The storm tide event from 1906 is the 13[th] highest for Cuxhaven since the start of recording in 1843 until now.

Although the event is important for the design and dimensioning of coastal defences (e.g. Streicher et al. 2015), knowledge about its details is still limited, and little is known about potential impacts such an event might have nowadays when hitting the present coastline.

To improve the data basis for the design of protection and optimization of forecasting procedures, Brecht and Frank (2015) and Streicher et al. (2015) constructed wind and water level fields, respectively, from several severe storm events between 1962 and 2011. While they acknowledged the importance of the 1906 event, it was still excluded in their analyses because of the lack of sufficiently reliable atmospheric data. While some historical weather maps and tide charts are available for this storm tide, no further attempts were made so far to simulate this event with numerical models or to assess potential impacts on the coast.

More recently, new sources became available from which atmospheric data needed to simulate the 1906 storm tide could be derived. First, century-long atmospheric reanalysis datasets were developed such as the European Reanalysis of Global Climate Observations (ERA-CLIM, Poli et al., 2016, Laloyaux et al., 2018) or the Twentieth Century Reanalysis (20CR, Compo et al., 2011, Slivinski et al., 2019). Second, Rosenhagen and Bork (2009) developed a semi-empirical approach using pressure data from archived historic and formerly non-digitized data to manually generate synoptic weather maps, from which pressure, geostrophic and near-surface marine wind fields are subsequently derived. Using this approach, they could, for example, reconstruct the very high water levels in the southwestern Baltic Sea that occurred during a storm event in November 1872. For this event, Feuchter et al. (2013) compared the pressure and wind fields from Rosenhagen and Bork (2009) with those from the Twentieth Century Reanalysis Project. They found substantial differences and concluded that the differences were mainly due to the different number of sea level pressure observations used in both approaches. Compared to the reanalysis, Rosenhagen and Bork (2009) used additional, newly digitized sea level pressure data available to them, where one of the additional pressure stations turned out to be essential for enhancing pressure gradients and wind speeds over the Baltic Sea.

In the following, we used both, wind and pressure fields from reanalyses data and the approach of Rosenhagen and Bork (2009) to simulate and assess the 1906 storm tide using a hydrodynamic model of the North Sea and the adjacent northeastern Atlantic. The objectives were (i) to explore the extent to which atmospheric data from the early years of century-long reanalysis datasets can be used to reconstruct the historical 1906 storm tide, (ii) to investigate to what extent wind fields from the semi-empirical approach of Rosenhagen and Bork (2009) can complement the reanalysis data by exploiting additionally archived but not yet digitized weather data, and (iii) to estimate if the height of this extreme event could have been higher, when the storm surge had hit the coast at a different phase of the astronomical tide.

The manuscript is structured as follows. In section 2, we describe the data and methods used. This comprises a description of the historic tide gauge data available, the additional weather data digitized for using the Rosenhagen and Borg (2009) approach together with a short description of the approach itself, and the various reanalyses products available for the hydrodynamic simulations. Also in section 2 the setup of the hydrodynamic tide-surge model and the experiments carried out to explore the potential amplification of the event are described. Results are presented in section 3 and comprise the description of the synoptic weather situation during the 1906 storm tide, a comparison of wind and pressure fields from the different reanalysis products and the semi-empirical approach, and a comparison of water levels simulated using different

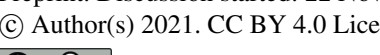


atmospheric forcings with the avaible observations. Also, the results of the amplification experiments are presented and discussed. The workflow and the experimental design is shown in Figure 2. Finally, the results are summarized and discussed in section 4.

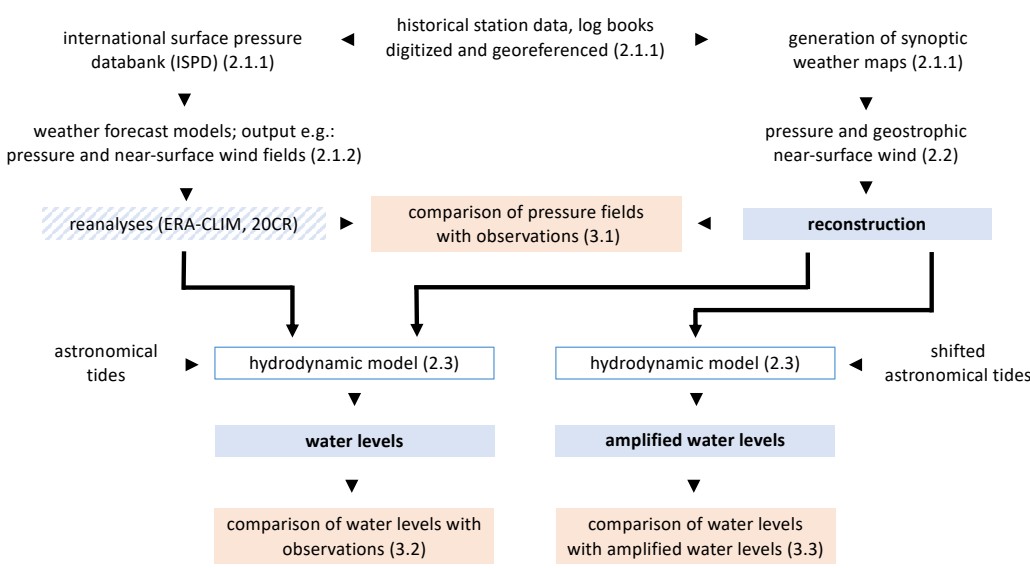

Figure 2: Workflow with numbering the corresponding chapters of this paper and the experimental design.

## 2 Data and Methods

### 2.1.1 Historical water level and weather data

For comparison with the simulated water levels, we used observations of high and low waters from the two tide gauges, Cuxhaven and Norderney (Figure 1), for which some data for the 1906 storm tide are available. These tide gauges have a long-term observation history: for Cuxhaven, a continuously time-series since 1843 (Deutsches Gewässerkundliches Jahrbuch (DGJ), 2014) and for Norderney since 1901 (pers. Comm. Cordula Berkenbrink) exist.

The Deutscher Wetterdienst (DWD, Germany's National Meteorological Service)  maintains a comprehensive archive of
historic weather data, substantial parts of which are still lacking digitization (weather report by the Kaiserliche Marine, Fig. 3). This also includes data of the 1906 storm tide event along the German North Sea coast and Europe (Figure 1).  For this study, twice-daily surface pressure data from more than one hundred stations for 1-15 March 1906 were digitized from this archive. These were subsequently used to derive wind fields using the Rosenhagen and Bork (2009) approach (section 2.3).

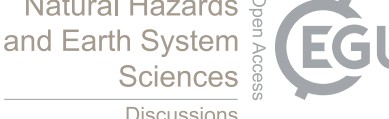

Figure 3: Example of a weather report for 12 March 1906 from the DWD archive from which pressure data were digitized and used to derive synoptic weather charts.
The location of these additional stations is shown in Figure 4 in comparison to stations available in the International Surface
Pressure Databank (ISPD) (Cram et al., 2015) which was used to construct the 20th-century reanalyses. The new pressure
data are provided to the ISPD, in which oberserved global historical pressure data are collected. Most of the new digitized
station data are in Central Europe. For comparison, we used recently digitized observations from so-called signal stations
located at Norderney, Helgoland, and Neuwerk (Figure 1) (Wagner et al., 2016).

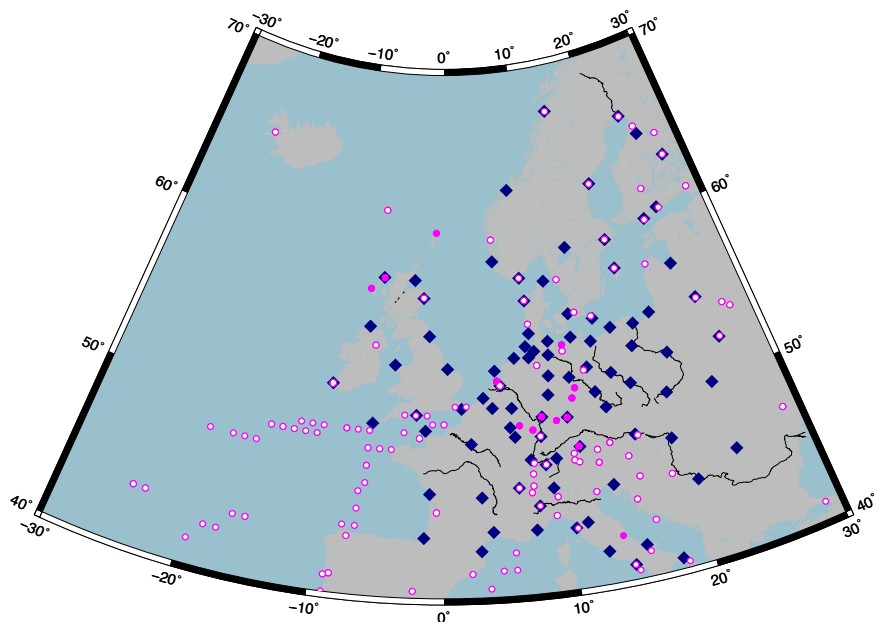

Figure 4: New digitized observation stations by the Deutscher Wetterdienst (blue diamond) and assimilated pressure data
from the ISPDv3 (white) in e.g. 20CRv2c, ERA-20C and CERA-20C and ISPDv4 (magenta) in 20CRv3. ISPDv4 is an
extension of ISPDv3 data and therefore they share the majority of stations.

### 2.1.2 Century reanalyses data

As atmospheric forcing for the hydrodynamic tide-surge model, we used datasets of sea level pressure and 10m-wind vectors
from the 20CR (Compo et al.,2011, Slivinski et al., 2019) and the ERA-CLIM projects (Poli et al., 2016, Laloyaux et al.,
2018), (Table 2). In both projects, the corresponding models assimilate station surface pressure data. The 20CR-project
further assimilates prescribed sea surface temperature and sea ice distributions while in ERA-CLIM marine wind (Poli et al.,
2016) and in ERA-CLIM2 ocean temperature and salinity profiles are additionally assimilated (Laloyaux et al., 2018).
From the 20CR-project both, the data from version 2c (20CRv2c) with 56 ensemble members and the updated version 3
(20CRv3) with 80 ensemble members were used. The main difference between both versions is the improved NCEP Global
Forecast System in 20CRv3 which has a higher resolution in both, time and space (Table 2). Moreover, the number of
assimilated surface pressure data increased, and data assimilation was updated including an adaptive inflation algorithm
(Slivinski et al., 2019). As a consequence, ensemble variability of sea surface pressure in 20CRv3 increased compared to
20CRv2c (Slivinski et al., 2019).





| Reanalysis | Shortcut | Ensemble member | Starting year | Spatial resolution | Temporal resolution |
|---|---|---|---|---|---|
| ERA-CLIM ERA-20C | ERA-20C | 1 | 1900 | ~ 210 km | 6/3-hourly |
| ERA-CLIM2 CERA-20C | CERA-20C | 10 | 1900 | 125 km | 3-hourly |
| 20th Century Reanalysis Project version 2c | 20CRv2c | 56 | 1851 | 2°x2° | 6/3-hourly |
| 20th Century Reanalysis Project version 3 | 20CRv3 | 80 | 1836 | 1°x1° | 3-hourly |

5   Table 2: Applied datasets of reanalysis forcing by ERA-CLIM projects and Twentieth Century projects.

From the ERA-CLIM projects, the datasets from ERA-20C (1 ensemble member) and CERA-20C (10 ensemble members) were used. CERA-20C based on the experiences of ERA-20C and extended by ocean and sea ice data (Laloyaux et al., 2018).

In total, wind and surface pressure fields from 147 different realizations of the March 1906 storm were thus available and used to produce an ensemble of storm surges and water levels during this event.

**2.2 Reconstruction of pressure and wind fields using the manual synoptic approach**

To exploit potential benefits of the additionally available non-digitized surface pressure data a 148th realization was created using the approach of Rosenhagen and Bork (2009). Here data from the additional stations (Figure 4) were digitized and these georeferenced data were plotted on twice-daily maps from which an experienced weather forecaster produced synoptic charts by manually drawing isobar lines (Figure 5) for the north-eastern part of the North Atlantic and Europe.
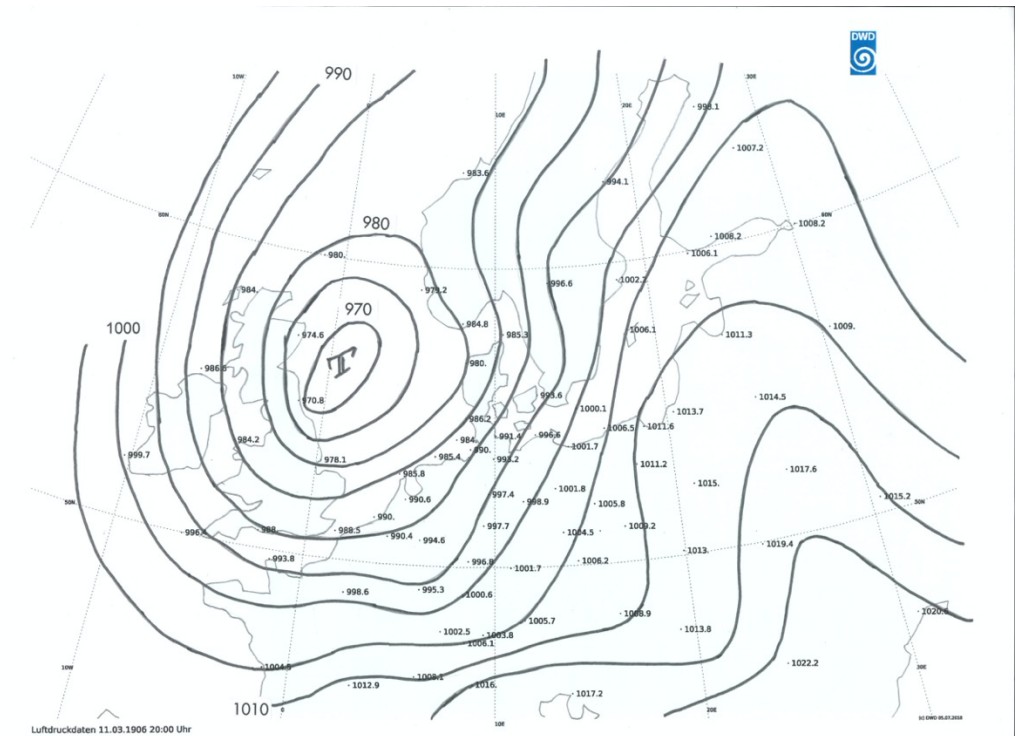

Figure 5: Example of a weather map derived from historic digitized pressure observations. The observed station values in hPa are shown together with isobars as drawn by experienced weather forecasters. The example shows the synoptic situation at 20:00CET 11 March 1906.

These synoptic charts were then re-digitized on a regular grid with a spacing of 0.07° from which then pressure gradients and geostrophic winds speeds (G) were computed on a 0.5° grid (Figure 6). To obtain near-surface marine wind speeds (U) needed to drive the hydrodynamic model, a simple parameterization of the boundary layer developed by Hasse (1974) was

10      subsequently applied. The parametrization was originally developed from observations taken at islands and lightships in the German Bight and nearby weather stations, and describes the near-surface marine wind speeds U as a function of the geostrophic wind G:

$$U = a * G + b$$

where

$$a = 0.54 - 0.012 * \Delta T$$
$$b = 1.68 - 0.105 * \Delta T$$

Here $\Delta T$ denotes the air-sea temperature difference and the parameterization was originally developed for $\Delta T$ ranging between -5 and 2.5K. Wind direction is subsequently derived from the geostrophic wind speed components.

In the following, this manual synoptic reconstruction is mentioned as "reconstruction".

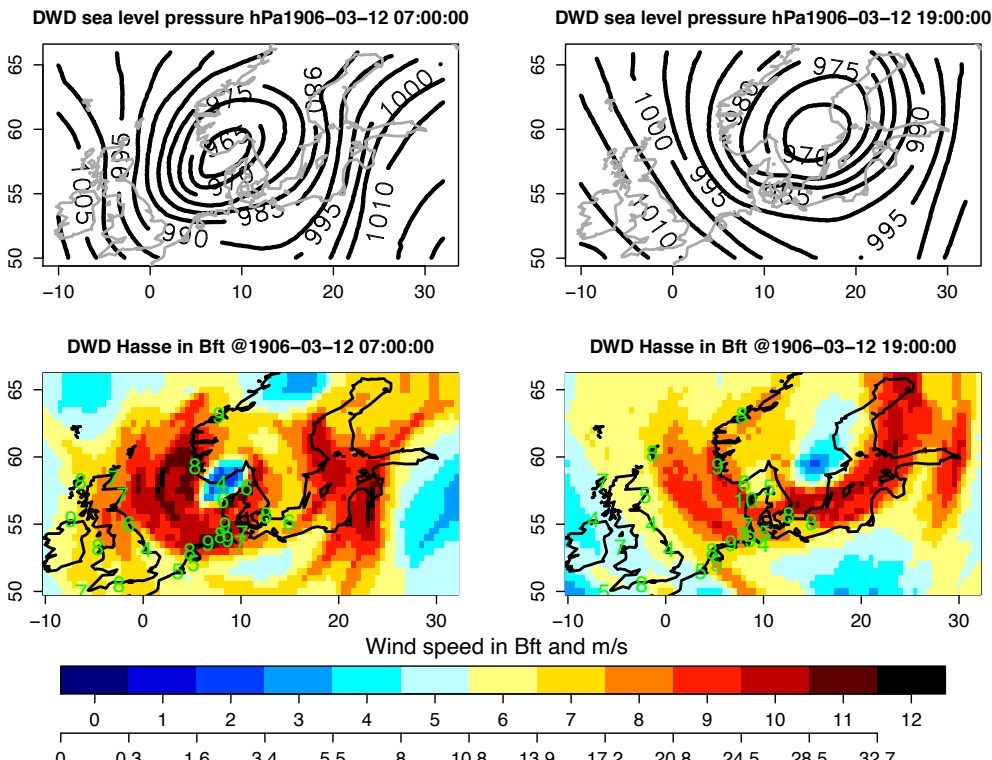

Figure 6: (Top) Sea level pressure maps (hPa) reconstructed from historic digitized weather data for 07:00 UTC 12 March 1906 (left) and 19:00 UTC 12 March 1906 (right). (Bottom) Corresponding near-surface wind speeds in Beaufort and ms⁻¹ (colour scale) derived from the Hasse (1974) parametrization. In addition, available reports from some coastal stations in Beaufort are shown (green).

### 2.3 Tide-surge model and experiments

Near-surface marine wind and pressure fields from the 148 realizations of the March 1906 storm were used to simulate water levels with the hydrodynamic tide-surge model TRIM-NP (Kapitza, 2008). TRIM stands for Tidal Residual and Intertidal

Mudflat model and represents a three-dimensional finite-difference model originally developed by Casulli and Cattani (1994) and later nested and parallelized (-NP) by Kapitza (2008). The model was extensively tested (Pätsch et al, 2017), validated, and used, for example, to hindcast tide-surge levels over decades of years (Gaslikova and Weisse, 2013), to simulate climate change projections (Gaslikova et al., 2013) or to analyse currents for drift simulations (Callies et al., 2011). In our case, the model was run in a three-level nested set-up with spatial grid sizes of 12.6 km × 12.6 km for grid 1, 6.4 km ×

6.4 km, 3.2 km × 3.2 km, and 1.6 km × 1.6 km for grid 4. In this set-up at the coarsest grid size, the model domain covers the area from 20°W to 30°E and from 42°N to 65°N. The higher resolution grids are located in the southern North Sea (Figure 1).

For all experiments, TRIM-NP was used in barotropic mode and was run for the period from 01 February – 15 March 1906.

Only for the reconstruction, the simulation started on 01 March 1906 because of data availability. Astronomical tides were included using data from the FES2004 global tide model (Lyard et al., 2006) as lateral boundary conditions along the

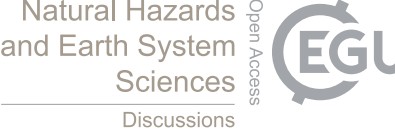

western and northern open boundaries of the model domain with the largest grid spacing. In addition, a tide-only simulation was performed to allow for the computation of surges from the full simulations.

Several sensitivity experiments were performed in which possible effects of the timing of the storm relative to the astronomical tide were investigated. For these experiments, lateral boundary conditions (tides) were shifted forwards or
backwards in time by one-hour steps. The objective was to investigate, whether or not such small shifts (which could be due to slight differences in the timing of the storm) would potentially have led to higher water levels which would have severe consequences for impacts.

For all simulations, model output was stored every hour for analyses.

### 3. Results

#### 3.1 The atmospheric situation 11-13 March 1906 and comparison of the atmospheric datasets

On 11 March, a low-pressure system with a core pressure of 970 hPa was located east of Scotland (Figure 5).  The pressure system moved eastward across the central North Sea and the Skagerrak to Sweden. Together with a high-pressure system
over the Bay of Biscay strong pressure gradients over the North Sea developed, leading to strong north-westerly winds over large parts of the North Sea (Figure 6). Based on the reconstruction, wind speeds generally exceeded 8 Beaufort (Bft) (> 17.2 ms$^{-1}$) and in some regions 10 Bft  (> 24.5 ms$^{-1}$) in the morning of 12 March 1906 over large parts of the North Sea (Figure 6). In the evening, the wind speeds were still higher than 8 Bft over the eastern central North Sea. Over the day, the prevailing wind direction changed from southwest to north-northwest. A comparison between the near-surface marine wind
speeds derived by the Hasse (1974) parametrization and observed wind speeds available at some coastal stations indicates that reconstructed wind speeds were somewhat higher along the East coast of Great Britain, but close to the observations along the coasts of Germany and Denmark (Figure 6). Over the western Baltic Sea, reconstructed wind speeds overestimate observations to some extent possibly due to the effect of the windward land which is not accounted for in the wind speed parameterization.  However, for the computation of surges in the North Sea, this overestimation is not relevant.


A comparison of wind speed time series from the different reanalyses products, the reconstruction, and observations from the signal stations Norderney, Helgoland, Neuwerk, and the weather station Cuxhaven for 12 March 1906 is shown in Figure 7. Generally, reported wind speeds were 9 Beaufort or higher for the coastal stations and varied between 7 and 11 Beaufort for the island of Helgoland. These observations and the wind speeds from the reconstruction are comparable. Wind speeds from
the ERA-CLIM products are generally lower than the observations. For most of the 20CR datasets, wind speeds are also lower than observed, but a few single ensemble members exist that show wind speeds comparable to the observations. The ensemble members from 20CRv3 provided both, the lowest and highest wind speeds for this storm event.




## Wind speed 10m 12.03.1906

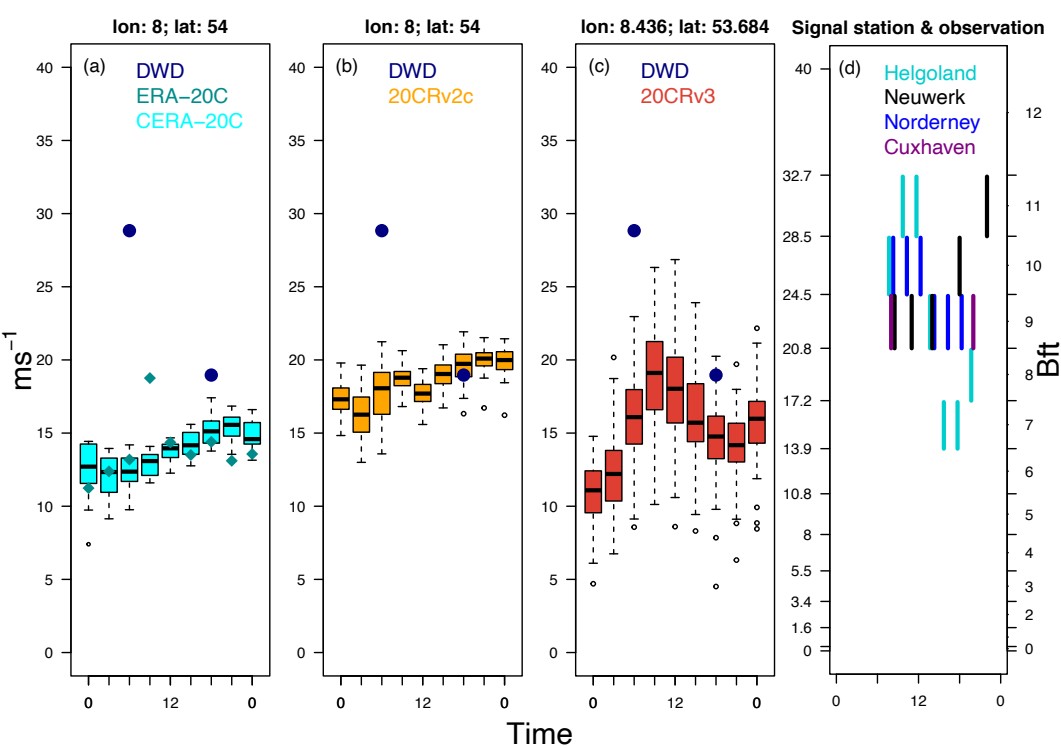

Figure 7: Wind speed in the German Bight on 12 March 1906. Data from the reconstruction from the DWD-data (blue bullets) in the panel (a)-(c), (a) wind speed from ERA-20C (dark cyan diamonds) and CERA-20C (cyan); (b) wind speed from 20CRv2c (orange) and (c) 20CRv3 (grenadine) in ms$^{-1}$. Data in their specific time resolution from the signal stations

Helgoland (cyan), Neuwerk (black) and Norderney (blue) and the weather station Cuxhaven (purple) is shown in the panel (d) in ms$^{-1}$ (left axis) and Bft (right panel) are shown for 12 March 1906.

An essential part of both the reanalyses and the reconstruction is how near-surface marine wind speeds are derived from

10 pressure fields. To assess potential uncertainties we also computed geostrophic wind speeds from the reanalyses products and compared them with those from the manual reconstruction for the Central North Sea  (Figure 8). Generally, all reanalyses products show the same daily cycle with maximum geostrophic wind speeds around noon on 12 March 1906. Variability between ensemble members is largest for 20CRv3 and smallest for 20CRv2c. The twice-daily values from the reconstruction are plotted in addition and are mostly close to those derived from the reanalyses. This indicates that surface

pressure fields between all products share major similarities.





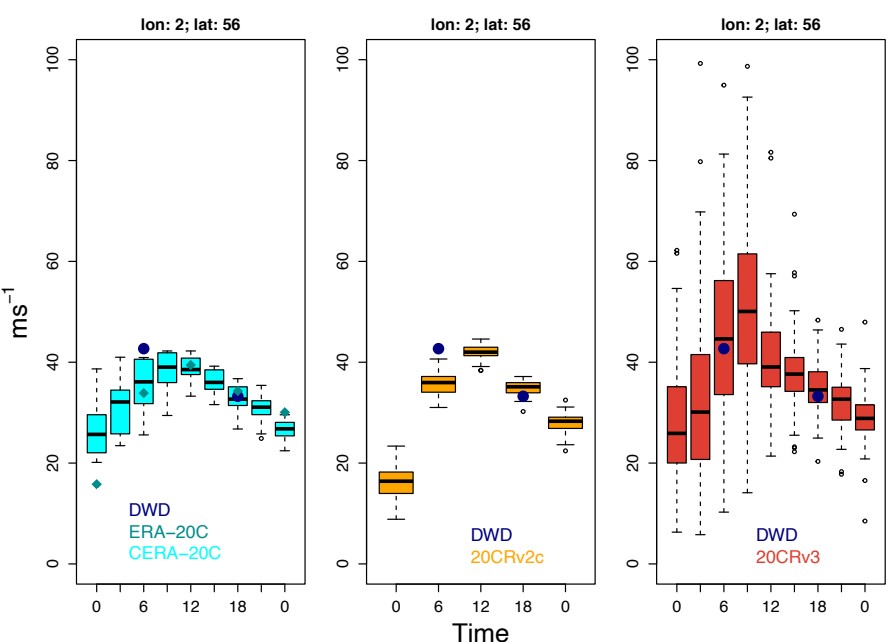

Figure 8: Geostrophic wind speed calculated from pressure fields of the particular reanalyses and reconstruction (dark blue dots). On the left side calculated geostrophic wind speed for ERA-20C (dark cyan) and CERA-20C (cyan), 20CRv2c (orange) in the middle and 20CRv3 (grenadine) on the right side.

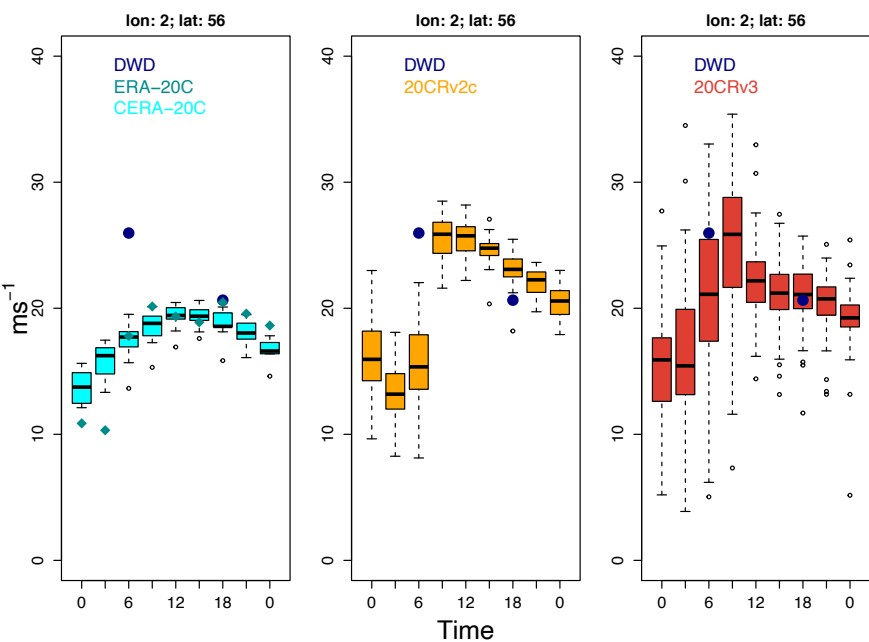

Figure 9: 10m wind speed of the respective reanalyses datasets and the calculated wind speed of the reconstruction (dark blue dots). On the left side 10m wind speed datasets from the ERA-20C and CERA-20C products, in the middle from 20CRv2c (orange) and on the right side from 20CRv3 (grenadine).





However, when near-surface marine wind speeds are considered, differences are much larger (Figure 9). In the morning, wind speeds from the ERA-CLIM products and 20CRv2c are now smaller than those from the reconstruction while for 20CRv3 singles members show the lowest and highest wind speeds.This indicates that despite similarities in the pressure

fields, the way near-surface marine wind speeds were calculated can have a substantial influence on the estimated wind field and introduce additional uncertainty when reconstructing water levels. This may include effects from boundary layer parameterizations in the reanalyses, for example, the way ocean waves were accounted for.

Figure 10 shows a comparison of the ensemble mean surface pressure fields derived from 20CRv3 and the reconstruction. Accounting for the fact, that the point in time of the observation differs from that of the model output, the similarity between

10 both fields is high, especially over the North Sea and regions with high data density.

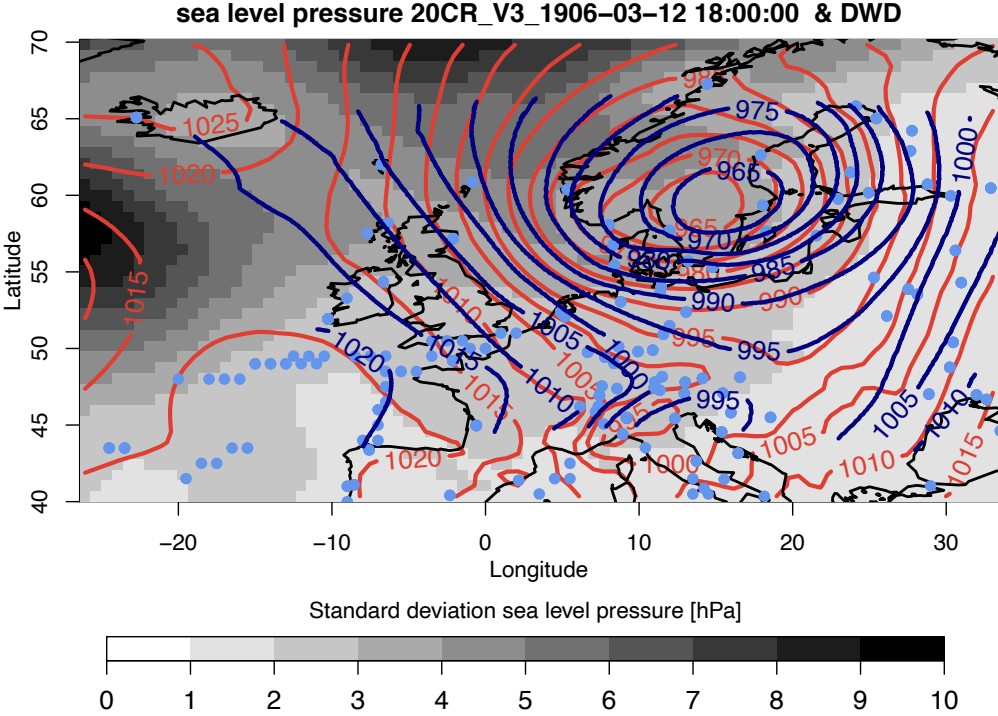

Figure 10: Pressure fields from DWD (dark blue, 19 UTC) and ensemble mean of pressure from 20CRv3 (grenadine) for 12 March 1906, 18 UTC. The standard deviation of sea level pressure of the 80 ensemble members for each grid box in hPa. The assimilated data from 12 March for the 20CRv3 dataset are marked in light blue.

Two inferences can be made: First, the similarity between the pressure charts suggests that the additional digitized data did not add value to the simulation of the wind fields compared to the used 20CRv3 reanalysis, because most of the new data are located in Central Europe. Second, in regions with a high density of assimilated data, ensemble variability in the reanalysis is substantially smaller compared to areas where no such data exists. The similarity supports the reliability of the reanalysis for

this event adding some confidence in the available wind and pressure fields.

### 3.2 Ensemble tide-surge simulations of the 1906 event

The 148 realizations of atmospheric conditions for the storm tide in March 1906 were used to generate an ensemble of corresponding tide-surge simulations for the event. Figure 11 shows a comparison of the observed and the modelled water levels at the two tide gauges at Norderney and Cuxhaven. Generally, low water levels and tidal ranges are underestimated in

all simulations. To some extent this might be a consequence of the present-day bathymetry used for the simulations which not necessarily represents the conditions in 1906. No bathymetric data were available for this time. Also, higher spatial resolution near the coast may be needed to better represent the details.

High water levels observed at the beginning of the simulation period are overestimated by the tide-surge simulation using wind and pressure fields from the reconstruction and the reanalyses ensemble members, but they are closer to the

observation compared to the low water level. The very severe conditions around 12 and 13 March are best reproduced using atmospheric fields of some of the 20CRv3 members, especially ensemble member 34 is close to the measured water levels at the Norderney gauge. The tide-surge simulations driven by 20CRv2c and the  reconstruction produce somewhat lower extremes while these are substantially underestimated when atmospheric forcing from the ERA-CLIM products is used.

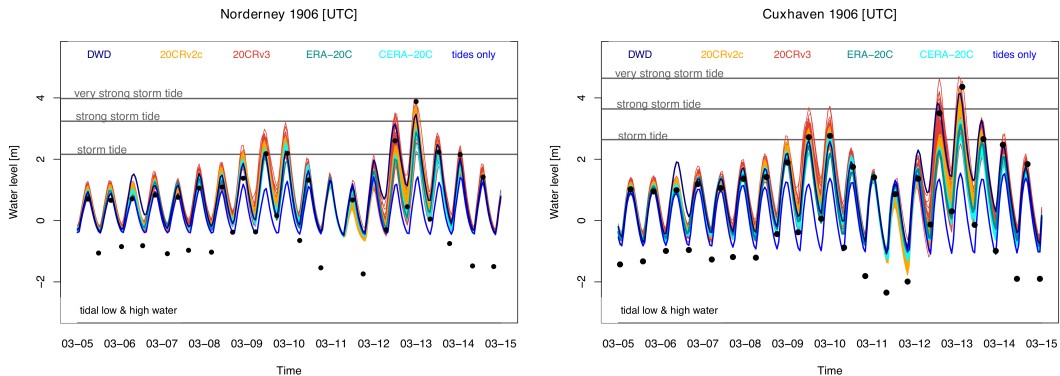

Figure 11: Observation (black dots) and simulation of water levels for the tide gauges Norderney and Cuxhaven. Water levels (tide and surge) forced by the manual synoptic reconstruction (dark blue) and by the reanalysis datasets from ERA-20C (dark cyan), CERA (cyan), 20CRv2c (orange) and 20CRv3 (grenadine). In addition, results from a tide only simulation (light blue) and in grey the storm tide criteria after DIN standard 4049. All results are in metres above Normaal Amsterdams Peil (NAP).

For both locations, the German DIN 4049-3 (Deutsches Institut für Normung e.V) defines a storm tide as an event that on average occurs between ten times a year and once every two years. Strong and very strong storm tides describe events that occur on average once every 2-20 and less than once every 20 years, respectively. Based on this classification and the data that became available over the past more than 100 years the storm tide of March 1906 would nowadays be qualified as a

strong storm tide, but close to the threshold for very severe cases (Figure 11). Only the wind fields in some of the atmospheric data from the 20CR reanalyses and the reconstruction are sufficiently strong to generate comparable water levels in our tide-surge simulations.

The spatial distribution of the maximum water levels that occurred during the event is shown in Figure 12. These water levels originate from the tide-surge simulation in which wind and pressure fields from the realization 34 of the 20CRv3





reanalysis were used. It can be inferred that the event affected large parts of the southern North Sea coastline. For five selected locations along the coasts of the Netherlands, Germany and Denmark the maximum simulated water levels exceed mean astronomical high waters by about 1.9 (location A) up to 3.4 m (location C). For the selected locations, these mean high waters were derived from time series of simulated astronomical tides from February to 15 March 1906.

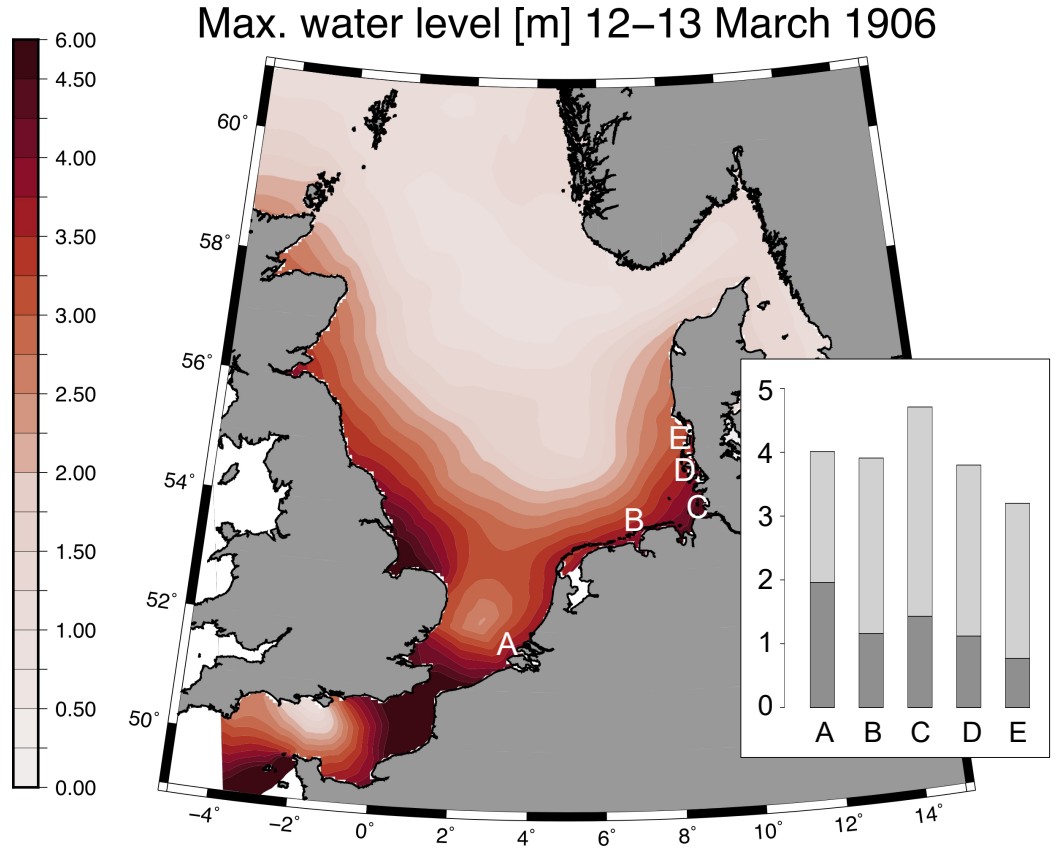

Figure 12: Maximum water levels in the North Sea during the event based on the tide-surge simulation forced by wind and pressure fields from the realization 34 of the 20CRv3 reanalysis. The insert shows bars for the simulated maximum water levels (light grey) compared to simulated astronomical high water levels (dark grey) averaged from 6 February to 15 March 1906 in metres above Normaal Amsterdams Peil (NAP) for five selected locations for which the positions are presented in the North Sea domain by the white letters.

### 3.3 Sensitivity experiments

While we were in principle able to reproduce observed extremes during the March 1906 storm tide along the German North

15   Sea coast using some of the reanalysed or the manually reconstructed atmospheric fields, the question remains, on whether or not the extreme water levels could have been further amplified by small changes in the met-ocean setting that could have occurred by chance. Such a modification could be, for example, a small delay (or the opposite) in the arrival of the storm peak. This could have an impact on extremes because of changes relative to the timing of the astronomical tide and tide-





surge interaction. Because of the large tidal amplitudes relative to the flat area, the timing of the storm and the tide will have an immediate effect on total water levels which may be complicated by effects from non-linear tide-surge interaction (Horsburgh and Wilson, 2007).

The most obvious difference would occur when the storm occurs during neap or spring tide. For Norderney, exemplarily,

5    differences between astronomical neap and spring tide can be in the order of 50 cm. The storm tide on 12-13 March 1906 occurred during spring tide so that further increases can be expected only for different time shifts between the astronomical tide and the storm maximum. To assess potential effects from smaller time shifts and the diurnal inequality, several sensitivity experiments were performed in which the phase of the astronomical tide was shifted hourly by several hours in both directions to mimick earlier/later arrivals of the storm peak at the coast. Results are shown for Norderney and Cuxhaven

10   (Figure 13). For this experiment, only the wind fields from the reconstruction are used.

The experiments show for Norderney and Cuxhaven only differences of a few centimetres between the members. For Cuxhaven, the simulation based on the reconstructed weather event without shifting the astronomical tide gives the highest water level. Thus, this storm tide event was almost the optimal convergence of high tide and surge.

### Norderney 1906 [UTC]

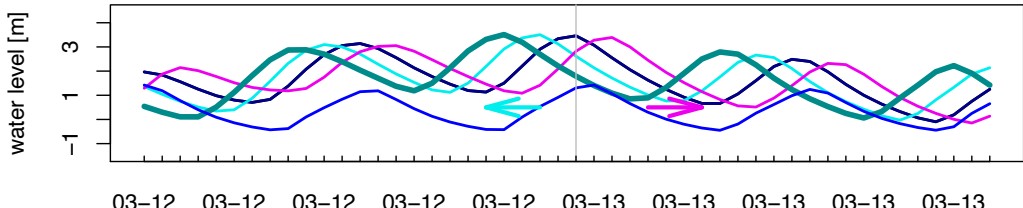

### Cuxhaven 1906 [UTC]

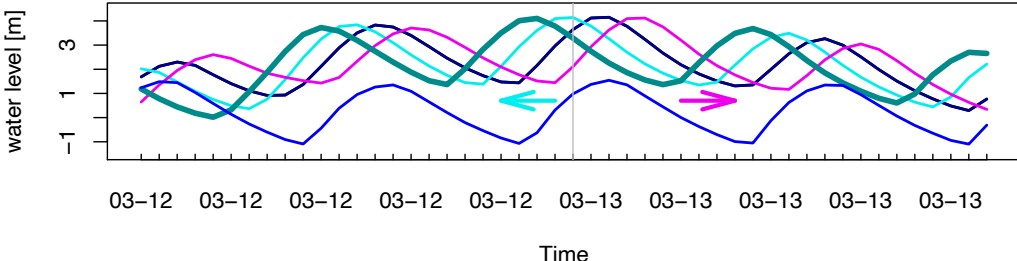

Figure 13: Reconstructed (dark blue) water levels at Norderney (top) and Cuxhaven (bottom) from a series of sensitivity experiments in which the timing of the astronomical tide (light blue) was shifted. Simulations in which the tide occurred in two-hour steps earlier/later relative to the storm are shown in cyan/magenta in metres above NAP.




### 4. Summary and Discussion

The water levels of a severe storm tide that occurred in the North Sea in March 1906 and affected large parts of the southern North Sea coast were reconstrucred. While the water levels from this event still provide some of the highest measured values at some tide gauges and are relevant for the design of coastal defences, numerical tide-surge simulations of the event were

lacking so far because of missing atmospheric data to drive such models. Using present-day bathymetric data, a tide-surge model was set-up that covered the North Sea and adjacent parts of the northeast North Atlantic. An ensemble of 147 realizations of the 1906 storm from different atmospheric reanalyses products together with a manual synoptic reconstruction that exploits additional observational data were used to drive the model.

It was tested, to what extent the observed water levels of the event could be reconstructed. It was found that some of the

atmospheric wind and pressure fields from some of the reanalyses realizations, in particular from 20CR, and the reconstruction were appropriate to simulate the observed water levels. Several sensitivity experiments were performed to explore the sensitivity of the extremes to the time difference between the astronomical tides and the arrival of the peak of the storm.

In particular, the results of this study suggest:

1. Atmospheric wind and pressure data from some of the available atmospheric reanalyses (in particular some realizations from 20CRv3 and 20CRv2c) as well as from the manual synoptic reconstruction are appropriate to reproduce observed water levels using a tide-surge model.
2. The manual synoptic reconstruction exploits additional available data, but does not add value to available

reanalyses in the sense that additional relevant synoptic features emerge. However, it helps in assessing the reliability of selected realizations from the reanalyses.
3. The storm peak hit the coast during a spring tide. Sensitivity experiments suggest, that the timing between the tide and the arrival of the storm peak was nearly optimal in a sense that no further substantial amplification could be achieved.


There are some points to consider in the interpretation of the results. The comparibillity of the tide-surge simulations with the little observational evidence available is limited for several reasons. No bathymetric data for the time of the event in 1906 are available. However, bathymetry is known to change substantially over years and decades in the area (e.g. Homeier et al., 2010, McBride et al. 2013). Furthermore, changes in the coastal protection systems, e.g. closing of the Ijsselmeer,

occurred. This will have an impact on the details of the development such as the exact timing or height of the water levels. Strictly speaking, the tide-surge simulations present the effects when the storm would hit the present coastline.

Concerning the mean sea level rise, the observed mean high waters in 1906 are expected to be lower. For comparison, observed annual mean high waters for 1906 are in the order of 15 cm lower for the gauges Norderney and Cuxhaven, than

the – astronomical high waters averaged over February to 15 March 1906 for locations B and C, which are somewhat more seaward than the gauges (Figure 12).

Moreover, the observations are subject to uncertainty as well. For comparison, we used observations of high and low waters from two tide gauges, Cuxhaven and Norderney, for which some data for the 1906 storm tide were available. For Norderney,

different sources comprise observations from 3.84 m to 4.30 m for this event in 1906 (time-series from the Wasserstraßen- und Schifffahrtsverwaltung des Bundes (WSV), communicated by Bundesanstalt für Gewässerkunde (BfG), DGJ 2014, page 42; Küste 33, page 21).



Finally, the spatial resolution of the tide-surge simulations is relatively coarse and details will change when further grid refinements are made. As bathymetric information for 1906 is unavailable and the objective was to assess the extent to which suitable atmospheric data for the event exist, we omitted further refinements. We suggest that the wind and atmospheric pressure fields from realization 34 of 20CRv3 and the reconstruction are suitable datasets to study the course

and potential impacts of the event. This can, for example, support risk assessment (Schaper et al., 2021), the basis for the design of defences, or the optimization of forecast procedures (Streicher et al. 2015).

**Data availability**

Meyer, E., Scholz, R., and Birger, T., Reconstruction of the 1906 Storm Tide in the German Bright using TRIM-NP, FES2004, and DWD weather data. World Data Center for Climate (WDCC) at

DKRZ, doi:10.26050/WDCC/storm_tide_1906_DWD_reconstruct, 2021.

Meyer, E., Reconstruction of the 1906 Storm Tide in the German Bright using TRIM-NP, FES2004, and ECMWF ERA-20C and CERA-20C reanalyses data. World Data Center for Climate (WDCC) at DKRZ. https://doi.org/10.26050/WDCC/storm_tide_1906_ERA-CLIM, 2021

Meyer, E., Reconstruction of the 1906 Storm Tide in the German Bright using TRIM-NP, FES2004, and NOAA-CIRES-DOE Twentieth Century Reanalysis (20CR) version 2c and 3. World Data Center for Climate (WDCC) at DKRZ. https://doi.org/10.26050/WDCC/storm_tide_1906_20CR, 2021.

The other data used in this paper are available from the authors upon request.

**Video supplement**

Meyer, Elke; Weisse, Ralf; Böttinger, Michael: The storm tide in March 1906. https://doi.org/10.5446/49529, 2020

**Author contribution**

The concept of this paper was jointly developed by all authors. All authors contributed to the preparation of the manuscript. The historical weather data were provided by BT. RS created the synoptic weather maps. Modelling work was carried out by

EM, analysis by EM, IG, and RW.

**Competing interests**

The authors declare that they have no conflict of interest.

**Acknowledgement**

The authors thank the staff of the Deutscher Wetterdienst (DWD) for digitizing historical weather data. We very much appreciate this great and time-consuming work. Sonja Ringelmann (DWD) has provided the weather maps. The Helmholtz Coastal Data Centre Team has digitized the weather maps. We thank Oliver Krüger for the help in using the ISPD datasets.

The high and low water level data for Norderney are from the Wasserstraßen- und Schifffahrtsverwaltung des Bundes (WSV), communicated by Bundesanstalt für Gewässerkunde (BfG) and the data for Cuxhaven are provided by the Bundesamt für Seeschifffahrt und Hydrographie.

Support for the Twentieth Century Reanalysis Project dataset is provided by the U.S. Department of Energy, Office of
Science Biological and Environmental Research (BER) program, by the National Oceanic and Atmospheric Administration Climate Program Office, and by the NOAA Physical Sciences Laboratory.



The work was carried out within the joint project EXTREMENESS (Extreme North Sea Storm Surges and Their Consequences) (funded by the German Federal Ministry of Education and Research(BMBF), Förderkennzeichen 03F0758)

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
