# Peer review of "Reconstruction of wind and surge of the 1906 storm tide at the German North Sea Coast"

_Natural Hazards and Earth System Sciences, 2021_

## Author Comment (AC3)

| Emden | | Norderney | | Wilhelmshaven | | Cuxhaven | |
|---|---|---|---|---|---|---|---|
| Lon: 7.2° | Lat: 53.3° | Lon: 7.1° | Lat: 53.7° | Lon: 8.1° | Lat: 53.5° | Lon: 8.7° | Lat: 53.9° |
| HW [m NAP] | Date | HW [m NAP] | Date | HW [m NAP] | Date | HW [m NAP] | Date |
| 5.18 | 1906-03-13 | 4.09 | 1962-02-16 | 5.18 | 1962-02-16 | 5.1 | 1976-01-03 |
| 5.17 | 2006-11-01 | 4.07 | 2013-12-06 | 5.08 | 2013-12-06 | : | : |
| 5.12 | 1944-02-04 | 3.95 | 1906-03-13 | 5.06 | 1906-03-13 | 4.36 | 1906-03-13 |

Table1: Observed high water (HW) levels above Normaal Amsterdams Peil (NAP) for selected tide gauges in the German Bight (Deutsches Gewässerkundliches Jahrbuch, 2013). The first station is located in the Ems estuary, Norderney is an East-Frisian Island, the gauge of Wilhelmshaven is situated in the Jade Bay, and the Cuxhaven gauge in the Elbe estuary. The storm tide event from 1906 is the 13[th] highest for Cuxhaven since the start of recording in 1843 until now.

---

## Author Response (AR1)

*We would like to thank the two reviewers for taking the time to review our manuscript and for their valuable comments and suggestions to improve our manuscript. We have written our responses below their points and how we addressed the individual issues raised by the reviewers in the revised manuscript (in orange).*

Reviewer 1:

The paper describes the numerical modelling of an extreme storm surge in 1906 in the North Sea using the atmospheric forcing from various reanalysis products and from a reconstruction based on station observations published in contemporary weather reports. The studied event is among the most severe storm surges that affected the German North Sea coast and, therefore, it deserves great attention.

The paper objectives appear clear but the part of the methodology is very questionable, particularly the manual reconstruction of pressure fields. Moreover, the text is often unclear.

Due to several problems, that are listed below, the work requires a major revision to make the paper suitable for publication.

Major remarks

1) Manual reconstruction of the pressure field.

Why was it done when many objective techniques are available?

At page 8, line 7, the authors say that the synoptic maps obtained manually were digitised on a 0.07° grid. So, why didn't the authors interpolate the weather report data onto such a grid directly?

I can see at least two critical issues related to a manual reconstruction. The first is that an objective approach allows to estimate the reconstruction error, which is clearly not possible from a manual reconstruction. The second is that, at least in principle, other scientists should be able to reproduce the results described in the paper and this is only possible if the reconstruction procedure is described mathematically, or, at least, if a standard software is used

*We agree that a manual analysis of the air pressure field does not allow an estimation of the error and is only partially reproducible. Despite that we decided to use the manual analysis because there exist several advantages that do not count in automatic procedures:*
*The approach in this paper was also used by Rosenhagen and Bork (2009), on whose work we rely.*
*Daily manual analysis of the surface air pressure was done in the operational forecasting office from 1876-2015 in the Deutsche Seewarte Hamburg and later in the Seewetteramt of the German Meteorological Service DWD. Two of the authors of the article have worked in operational shifts and contribute many years of experience in drawing surface pressure charts.*
*Compared to an automatic interpolation method, the forecaster will keep in mind the air pressure distribution of the previous weather maps for deriving the air pressure tendencies in the manual analysis. This can be used to improve the accurate location of the cyclone and the position of the isobars in areas with poor data coverage. Incorrect air pressure values can be reliably detected by an experienced meteorologist and these values are then not used further in the analysis*
*Furthermore, the distance between the isobars is linked to the wind speed and the curvature of the isobars, which cannot be considered with automatic methods, but in a manual analysis. While the manual analysis thus involves some degree of subjective assessment, the procedure is reproducible (as published in*

*Rosenhagen and Bork 2009) and the results are available and can be used by others for analysis or driving models and comparison.*

I also noted that, at least from the weather report in fig. 3, pressure used to be measured at different times between 7 and 8 CET, depending on the country. How did the manual reconstruction take into account different times? A 1-hour lag can be significant, particularly with regard to the simulations sensitivity to the astronomical tide shifts, discussed in sect. 3.3.

*The reports in Great Britain, Norway, Denmark, Belgium, the Netherlands, Sweden, Austria, and Germany were made at 8 a.m. CET, in the other countries it was 7 a.m. local time or CET. The air pressure data from 8 CET are therefore in the relevant area of the map. The different times were considered in the manual analyses.*

2) Page 13, lines 16-20. The text reads:

'Two inferences can be made: First, the similarity between the pressure charts suggests that the additional digitized data did not add value to the simulation of the wind fields compared to the used 20CRv3 reanalysis, because most of the new data are located in Central Europe. Second, in regions with a high density of assimilated data, ensemble variability in the reanalysis is substantially smaller compared to areas where no such data exists. The similarity supports the reliability of the reanalysis for this event adding some confidence in the available wind and pressure fields.'

What does 'similarity' mean? I understand that it is not a quantitative measure of how different the two charts are, but rather that it means that they look more or less the same. Then the conclusions are totally subjective.

At least for the purpose of wind reconstruction, the authors conclude that the weather report data do not seem to be useful. Therefore, considering the questionable way in which those data are used (point 1 above), it would be reasonable to drop everything connected with them.

Last but not least, I cannot agree that the reanalyses are reliable because they are similar to maps drawn by hand!

*We agree with the reviewer, that the comparison is subjective. We did this comparison to check if the patterns of the simulated isobars from the reanalyses are comparable with the weather situation, e.g. with the reconstruction. For long-ago historical storms, there may not be enough pressure data in key regions, so the storms may not be reconstructed in detail. For example:*

*Feuchter et al., 2013 investigated the 20CRv2 pressure data with the pressure data from a reconstruction by Rosenhagen and Bork (2009) for the storm event 13.11.1872 in the Baltic Sea. In this case, a missing pressure observation has changed the isobars for this storm in the reconstruction resulting in stronger gradients over the Bay of Lubeck compared with the reanalyses. Also, an investigation by Ed Hawkins et al. 2019 shows in a case study that added digitized data improved understanding of the severe storm from 1903 over the British Isles. In the 20CRv3-data these added data were not included and the reanalyses do not simulate a severe storm. In his talk '**Improving Atmospheric Reconstructions For Historical Extreme Events',** Ed Hawkins shows the added value of such missing historical data for the severe storm in 1903 (https://www.rmets.org/event/virtual-meeting-improving-atmospheric-reconstructions-historical-extreme-events)*

*For the severe storm 1906, there were data at the crucial regions already available and assimilated in the reanalyse model. With the reconstruction, we can verify the pressure pattern with the reanalysis model data.*

*We addressed the points/concerns raised by the reviewer and revised these points in the chapters "1 Introduction" [page 3, lines 15-31 in the marked-up manuscript version] and "2.2. Reconstruction of sea level pressure and wind fields using the manual synoptic approach" [page 7, lines 23-27 in the marked-up manuscript version] to make them clearer.*

*Hawkins et al., 2019, 'Hourly weather observations from the Scottish Highlands (1883–1904) rescued by volunteer citizen scientists', Geoscience Data, 6, 160, doi: 10.1002/gdj3.79*

3) 3.2 Ensemble simulations.

The authors say that 148 simulations were made (page 14, line 2). Only after magnification, fig. 11 reveals a large number of curves (all 148?), most of which cannot be discriminated from each other, causing the display to be almost useless. Besides improving the curve visibility (e.g. using larger figures), the authors should provide a synthetic assessment of their results, for instance by means of standard errors between simulations and observations. This would enable to specify to what extent the simulations succeed in reconstructing the storm surge: At the moment, the sentences at page 17, lines 9-10 ('It was tested to what extent … It was found that some …') are inconclusive.

*We split the figures to enhance the visibility. For the discussion it is important to make the most extreme water level visible. Therefore, we refrain from a synthetic assessment that will blur the individual results. [page 15- 16 in the marked-up manuscript version]*

4) Page 17, lines 16-21: Results 1 and 2. The text reads:

'1. Atmospheric wind and pressure data from some of the available atmospheric reanalyses (in particular some realizations from 20CRv3 and 20CRv2c) as well as from the manual synoptic reconstruction are appropriate to reproduce observed water levels using a tide-surge model.

2. The manual synoptic reconstruction exploits additional available data, but does not add value to available 20 reanalyses in the sense that additional relevant synoptic features emerge. However, it helps in assessing the reliability of selected realizations from the reanalyses.'

Concerning result no. 1, fig. 11 shows that only the times of high/low waters are reproduced well, and, with regard to this, note that the astronomical tide is dominant. By contrast, the considerable spread among the peaks heights, up to over 1 m on 13 March, raises the problem of uncertainty. The authors should explain what they mean by 'appropriate' by discussing what is considered to be a useful or acceptable uncertainty for a specific purpose, for instance coastal defence planning.

*We elaborated on these points in more detail in the chapter "4. Summary and Discussion" [page 19, lines 12-15 in the marked-up manuscript version]*

*Until now, it has not been possible to perform these studies due to the lack of atmospheric forcing data. With the reanalyses and our reconstruction, we show what options are possible. The range of water level simulations at the peak of the event is large, but all results would be physically plausible. These results can add value to the assessment of applied coastal protection by showing that it is possible to simulate this event and future comparable situations can be detected and actions can be taken earlier.*

Result no. 2 can certainly depend on the spatial data coverage, which favours Central Europe as explained by the authors at page 13, lines 16-17, but also on the availability of the atmospheric forcing only every 12

hours. For comparison, the reanalyses are available every 3 or 6 hours. This point should also be discussed.

*This point of the reviewer is discussed in more detail in the revised version. [page 19, lines17-19 in the marked-up manuscript version]*

Minor problems

1) Please make it clear when sea-level pressure and surface pressure are used. For instance, at page 4, line 17, in reference to the weather report of fig. 3 'surface pressure' is used, but pressure is actually reduced to the sea level (bottom of the figure). Similarly, the text at page 13, line 8 reads 'surface pressure', the caption of fig. 10 'pressure' and the legend of fig. 10 'sea level pressure'.

*Thank you for the hint. We have changed it to sea-level pressure where necessary.*

2) Page 16, line 11: I understand that the few centimetres differences refer to the peak heights. Please clarify this point. At a given time, differences can be as large as 1 m.

*Yes, the few centimetres difference refers to the peak heights. We clarified this in the text. [page 18, line 6 in the marked-up manuscript version]*

3) Figures are often hard to understand because they are full of information and colours, namely:

a) In fig. 6 the green labels in the bottom panels are difficult to see.

*We changed it to a better visible colour (black). [page 9 in the marked-up manuscript version]*

b) In figs. 7, 8 and 9 the complex symbols require a full description: mean, +/- STD, extremes … Moreover, what do the small circles represent?

*We added the explanation for the symbols. [page 11-13 in the marked-up manuscript version]*

*Caption: … In the boxplot, the box represents the interquartile range (IQR) and the median is given by the thick black line. The whiskers represent all data lying within 1.5 times the IQR while data outside this range are shown by circles.*

c) In fig. 13 what are the thick dark green curves? The caption misses this information. Moreover, the abscissa labels should show the hour while the month is not important.

*We complemented the caption. The dark cyan line is 4 hours earlier than the event. We added this information in the caption. Also, we changed the x-labels in hours. [page 18, lines 14-15 in the marked-up manuscript version]*

4) Page 17, lines 33-36: 'Concerning the mean sea level rise, the observed mean high waters in 1906 are expected to be lower. …'

The whole paragraph is unclear. In particular, the mean high waters in 1906 are expected to be lower than what and why? They are said to be observed (line 33). Please explain.

*We agree with the reviewer, that the mean sea level rise is not an issue in this paper and therefore we skipped this paragraph.*

5) Page 17, line 40: 'different sources …'

I do not understand if the reported measurements come from tide gauges or not. The 3.84 m height at Norderney seems to have been used in fig. 11, but what is the role of the 4.30 m height in the comparisons? Please explain.

*There is uncertainty in the observation of the peak height of the severe storm in 1906. In our investigation, we have found three different sources for the peak water level in the literature:*

*Wasserstraßen- und Schifffahrtsverwaltung des Bundes (WSV), communicated by Bundesanstalt für Gewässerkunde (BfG);*

> *observed peak height = 3.84 m*

*Deutsches Gewässerkundliches Jahrbuch (DGJ) 2014, page 42;*

> *observed peak height = 3.95 m*

*Die Küste 33, page 21;*

> *observed peak height = 4.30 m*

*All sources are trustworthy.*

*We extended the discussion on that in the revised manuscript. [page 19, lines 33-36 in the marked-up manuscript version]*

Reviewer 2:

recommendation: minor revision

General comments:

The manuscript deals with the reconstruction of a storm surge event occurred in 1906 in the North Sea. The article is well written, pleasant to read, and scientifically sound. I have only some minor comments that the authors should clarify before this manuscript can be published.

Specific comments:

1,18: potential amplification… I would not call it potential amplification but constructive superposition or similar

*We have thought about the argument but found "constructive superposition" not fully suitable. We propose to keep potential amplification, but we will add a clear definition of what we mean by this wording. The same wording is used in*

*Grabemann, I., Gaslikova, L., Brodhagen, T., and Rudolph, E.: Extreme storm tides in the German Bight (North Sea) and their potential for amplification, Nat. Hazards Earth Syst. Sci., 20, 1985–2000, doi:10.5194/nhess-20-1985-2020, 2020.*

*We added a short description in the text, [page 3, line 37 in the marked-up manuscript version]*

Table 1: what do the two dots ":" mean in the tide gauge of Cuxhaven? If they mean "some other events" and then the 13th highest of 1906, the date 2013-12-06 has to be eliminated and substituted by ":".

*The reviewer may be looking at the wrong column.*

*We have drawn some lines between the columns to clarify which HW and Date belong together [page 2, line 10 in the marked-up manuscript version].*

Fig. 2: the entry "amplified water levels" is misleading. You may consider shifted storm surge or similar.

*We changed it to "shifted storm tides" in figure 2. [page 4, line 9 in the marked-up manuscript version]*

9,15: missing information grid 2 and grid 3

*We complemented "6.4 km × 6.4 km for grid 2, 3.2 km × 3.2 km for grid 3" in the text. [page 9, line 15 in the marked-up manuscript version]*

9,17: is this a bi-directional coupling, or only one-directional? If it is one-directional, what are the consequences of not allowing the higher and better simulated water levels leaving the fine grid and modifying the boundary conditions that are imposed? Please discuss.

*This is a one-directional coupling. The differences are only a few centimetres for the total water level. The biggest differences are in the calculation of wind surge because the resolution of the bathymetry for grid 1 is low compared to the resolution of the fine grid 4.*

Fig 7 and 8: wind speeds from reconstruction are normally lower than 20 m/s, but geostrophic wind speeds can go up to 40 m/s and even higher. Can you explain this feature?

*First, figures 7 and 8 are for different locations. Figures 8 and 9 are for the same location. We intended to make the reanalyses comparable by calculating the geostrophic wind (figure 8). From figure 8 we could conclude that the median geostrophic wind speed is almost the same. Geostrophic wind represents a balance between pressure gradient and Coriolis force. Surface winds are generally weaker including friction, vertical stability, and other effects. A comparison is available, for example by the WASA Group (1998).*

*Moreover, the differences in 10m wind speed are higher between the individual models. The models for the reanalyses use different parameterizations for the calculation of the wind speed. Especially the products from ECMWF stand out due to the lower simulated wind speed in extreme situations, see also Cavaleri et al., 2019 for the ECMWF-forecast.*

*The WASA Group (1998). Changing Waves and Storms in the Northeast Atlantic?. Bulletin of the American Meteorological Society 79, 5, 741-760, available from: <https://doi.org/10.1175/1520-0477(1998)079<0741:CWASIT>2.0.CO;2>*

16,1-3: please also see Ferrarin et al., 2021 and Cavaleri et al., 2019 and 2020 for adding different components of the water level and the importance of timing for storm surges occurred in Venice

*Thank you for the references. We will consider them in the revised manuscript.*

We considered Cavaleri et al., 2020 in the text. [page 17, line 17 *in the marked-up manuscript version*]

Fig 13: it would be beneficial to also see only the storm surge component. If the plateau of the storm surge peak were quite flat, then a small shift of the timing would not have a big effect.

We do not fully agree.
The argument would hold in case of small or missing tide-surge interaction. In shallow water this is generally not the case. We therefore prefer to keep the original figures.

Here some references. The authors should feel free and not obliged to use these if they think they are appropriate.

*Thank you for the references. We will consider them in the revised manuscript.*

Ferrarin, C., Bajo, M., Benetazzo, A., Cavaleri, L., Chiggiato, J., Davison, S., Davolio, S., Lionello, P., OrliÄ, M., and Umgiesser, G. (2021). Local and large-scale controls of the exceptional Venice floods of November 2019. Progress in Oceanography, 197,102628. DOI:10.1016/j.pocean.2021.102628

Cavaleri, L., Bajo, M., Barbariol, F., Bastianini, M., Benetazzo, A., Bertotti, L., Chiggiato, J., Ferrarin, C., Trincardi, F., and Umgiesser, G. (2020). The 2019 Flooding of Venice and its implications for future predictions. Oceanography, 33(1), 42-49. DOI:10.5670/oceanog.2020.105

Cavaleri, L., Bajo, M., Barbariol, F., Bastianini, M., Benetazzo, A., Bertotti, L., Chiggiato, J., Davolio, S., Ferrarin, C., Magnusson, L., Papa, A., Pezzutto, P., Pomaro, A., and Umgiesser, G. (2019). The October 29, 2018 storm in Northern Italy — an exceptional event and its modeling. Progress in Oceanography, 178, 102178. DOI:10.1016/j.pocean.2019.102178

---

## Referee Report (RR1)

Review of the paper "Reconstruction of wind and surge of the 1906 storm tide at the German North Sea Coast" by Meyer et al. (revised version)

I thank the authors for considering my remarks, most of which were satisfactory addressed.
I want to make it clear that I am not questioning the expertise and competence of the people who produced the manually interpolated pressure fields. Probably, different forecasters would draw maps that are not too different from each other, and this would not be critical for the simulation of the 1906 event. I would just like to point out that the manual reconstruction cannot be considered as a proper reference (see my comment to No. 2, below), but rather one possible reconstruction.

My comments to the authors' replies follow. I only kept the relevant points and sentences of the authors' response document. Pages and lines are those of the "clean" text (not the marked-up version).

The paper can be published after taking into account these remarks.

(Note: At least from about page 14 the pages and lines mentioned in the answers do not match with the pdf.)

1) Manual reconstruction of the pressure field.

The first is that an objective
approach allows to estimate the reconstruction error, which is clearly not possible from a manual reconstruction.
*Compared to an automatic interpolation method, the forecaster will keep in mind the air pressure distribution of the previous weather maps for deriving the air pressure tendencies in the manual analysis.*
*This can be used to improve the accurate location of the cyclone and the position of the isobars in areas*
*with poor data coverage. Incorrect air pressure values can be reliably detected by an experienced meteorologist and these values are then not used further in the analysis*
*Furthermore, the distance between the isobars is linked to the wind speed and the curvature of the isobars, which cannot be considered with automatic methods, but in a manual analysis. While the manual*
*analysis thus involves some degree of subjective assessment, the procedure is reproducible (as published in*
*Rosenhagen and Bork 2009) and the results are available and can be used by others for analysis or driving*
*models and comparison.*
I also noted that, at least from the weather report in fig. 3, pressure used to be measured at different times between 7 and 8 CET, ...
*The reports in Great Britain, Norway, Denmark, Belgium, the Netherlands, Sweden, Austria, and Germany*
*were made at 8 a.m. CET, in the other countries it was 7 a.m. local time or CET. The air pressure data from*
*8 CET are therefore in the relevant area of the map. The different times were considered in the manual analyses.*

I mentioned an "objective method" of interpolation (for instance objective analysis, optimal interpolation), which does not mean "automatic" because it requires the user to provide appropriate covariance functions, based on the analysis of the observations. The interpolation can be made simultaneously in space and in time, thus using the information coming from previous and following synoptic observations.

2) Page 13, lines 16-20. The text reads:
'Two inferences can be made: First, the similarity between the pressure charts suggests that the additional digitized data did not add value to the simulation of the wind fields compared to the used 20CRv3 reanalysis, because most of the new data are located in Central Europe. Second, in regions with a high density of assimilated data, ensemble variability in the reanalysis is substantially smaller compared to areas where no such data exists. The similarity supports the reliability of the reanalysis for this event adding some confidence in the available wind and pressure fields.'
What does 'similarity' mean? I understand that it is not a quantitative measure of how different the two charts are, but rather that it means that they look more or less the same. Then the conclusions are totally subjective.
At least for the purpose of wind reconstruction, the authors conclude that the weather report data do not seem to be useful. Therefore, considering the questionable way in which those data are used (point 1 above), it would be reasonable to drop everything connected with them.
Last but not least, I cannot agree that the reanalyses are reliable because they are similar to maps drawn by hand!

*We agree with the reviewer, that the comparison is subjective. We did this comparison to check if the patterns of the simulated isobars from the reanalyses are comparable with the weather situation, e.g. with the reconstruction. For long-ago historical storms, there may not be enough pressure data in key regions, so the storms may not be reconstructed in detail. For example:*
*Feuchter et al., 2013 investigated the 20CRv2 pressure data with the pressure data from a reconstruction*
*by Rosenhagen and Bork (2009) for the storm event 13.11.1872 in the Baltic Sea. In this case, a missing*
*pressure observation has changed the isobars for this storm in the reconstruction resulting in stronger gradients over the Bay of Lubeck compared with the reanalyses. Also, an investigation by Ed Hawkins et al.*
*2019 shows in a case study that added digitized data improved understanding of the severe storm from 1903 over the British Isles. In the 20CRv3-data these added data were not included and the reanalyses do*
*not simulate a severe storm. In his talk '**Improving Atmospheric Reconstructions For Historical Extreme***
***Events',** Ed Hawkins shows the added value of such missing historical data for the severe storm in 1903*
*(https://www.rmets.org/event/virtual-meeting-improving-atmospheric-reconstructions-historicalextreme-events)*
*For the severe storm 1906, there were data at the crucial regions already available and assimilated in the*
*reanalyse model. With the reconstruction, we can verify the pressure pattern with the reanalysis model data.*
*We addressed the points/concerns raised by the reviewer and revised these points in the chapters "1 Introduction" [page 3, lines 15-31 in the marked-up manuscript version] and "2.2. Reconstruction of sea level pressure and wind fields using the manual synoptic approach" [page 7, lines 23-27 in the marked-up*
*manuscript version] to make them clearer.*
***Hawkins** et al., 2019, 'Hourly weather observations from the Scottish Highlands (1883–1904) rescued by*
*volunteer citizen scientists', Geoscience Data, 6, 160, doi: 10.1002/gdj3.79*

I fully agree that data archaeology is extremely valuable; the reanalyses themselves are produced using historical observations.
The piece of text I was referring to (now on page 13, lines 15-22) was extended but the authors did not answer the question about similarity, and the reanalyses are still considered reliable because they look similar to the manual reconstructions. The authors seem to consider the manual reconstruction as a reference but I think that scientific results should have an objective basis, when available.
I suggest to say that, because the maps look similar, the main features of the respective wind fields are consistent.

3) 3.2 Ensemble simulations.

*We split the figures to enhance the visibility. For the discussion it is important to make the most extreme water level visible. Therefore, we refrain from a synthetic assessment that will blur the individual results. [page 15- 16 in the marked-up manuscript version]*

Ok, but the figures are still very busy, also because of the relatively large tidal signal. What about showing the de-tided curves and data?

---

## Author Response (AR2)

Dear Editor and reviewer 1,

We thank the reviewer 1 for the helpful remarks. Our respective comments are in green.

Review of the paper "Reconstruction of wind and surge of the 1906 storm tide at the German North Sea Coast" by Meyer et al. (revised version)

I thank the authors for considering my remarks, most of which were satisfactory addressed. I want to make it clear that I am not questioning the expertise and competence of the people who produced the manually interpolated pressure fields. Probably, different forecasters would draw maps that are not too different from each other, and this would not be critical for the simulation of the 1906 event. I would just like to point out that the manual reconstruction cannot be considered as a proper reference (see my comment to No. 2, below), but rather one possible reconstruction.

My comments to the authors' replies follow. I only kept the relevant points and sentences of the authors' response document. Pages and lines are those of the "clean" text (not the marked-up version).

The paper can be published after taking into account these remarks.

(Note: At least from about page 14 the pages and lines mentioned in the answers do not match with the pdf.)

1) Manual reconstruction of the pressure field.

The first is that an objective
approach allows to estimate the reconstruction error, which is clearly not possible from a manual reconstruction.
Compared to an automatic interpolation method, the forecaster will keep in mind the air pressure distribution of the previous weather maps for deriving the air pressure tendencies in the manual analysis.
This can be used to improve the accurate location of the cyclone and the position of the isobars in areas
with poor data coverage. Incorrect air pressure values can be reliably detected by an experienced meteorologist and these values are then not used further in the analysis
Furthermore, the distance between the isobars is linked to the wind speed and the curvature of the isobars, which cannot be considered with automatic methods, but in a manual analysis. While the manual
analysis thus involves some degree of subjective assessment, the procedure is reproducible (as published in
Rosenhagen and Bork 2009) and the results are available and can be used by others for analysis or driving
models and comparison.
I also noted that, at least from the weather report in fig. 3, pressure used to be measured at different times between 7 and 8 CET, ...
The reports in Great Britain, Norway, Denmark, Belgium, the Netherlands, Sweden, Austria, and Germany
were made at 8 a.m. CET, in the other countries it was 7 a.m. local time or CET. The air

pressure data from
8 CET are therefore in the relevant area of the map. The different times were considered in the manual analyses.

I mentioned an "objective method" of interpolation (for instance objective analysis, optimal interpolation), which does not mean "automatic" because it requires the user to provide appropriate covariance functions, based on the analysis of the observations. The interpolation can be made simultaneously in space and in time, thus using the information coming from previous and following synoptic observations.

The reviewer is right, but the line of argumentation above still holds.

2) Page 13, lines 16-20. The text reads:
'Two inferences can be made: First, the similarity between the pressure charts suggests that the additional digitized data did not add value to the simulation of the wind fields compared to the used 20CRv3 reanalysis, because most of the new data are located in Central Europe. Second, in regions with a
high density of assimilated data, ensemble variability in the reanalysis is substantially smaller compared to areas where no such data exists. The similarity supports the reliability of the reanalysis for this event
adding some confidence in the available wind and pressure fields.'
What does 'similarity' mean? I understand that it is not a quantitative measure of how different the two
charts are, but rather that it means that they look more or less the same. Then the conclusions are totally subjective.
At least for the purpose of wind reconstruction, the authors conclude that the weather report data do not
seem to be useful. Therefore, considering the questionable way in which those data are used (point 1 above), it would be reasonable to drop everything connected with them.
Last but not least, I cannot agree that the reanalyses are reliable because they are similar to maps drawn
by hand!
We agree with the reviewer, that the comparison is subjective. We did this comparison to check if the patterns of the simulated isobars from the reanalyses are comparable with the weather situation, e.g. with the reconstruction. For long-ago historical storms, there may not be enough pressure data in key regions, so the storms may not be reconstructed in detail. For example:
Feuchter et al., 2013 investigated the 20CRv2 pressure data with the pressure data from a reconstruction
by Rosenhagen and Bork (2009) for the storm event 13.11.1872 in the Baltic Sea. In this case, a missing
pressure observation has changed the isobars for this storm in the reconstruction resulting in stronger gradients over the Bay of Lubeck compared with the reanalyses. Also, an investigation by Ed Hawkins et al.
2019 shows in a case study that added digitized data improved understanding of the severe storm from 1903 over the British Isles. In the 20CRv3-data these added data were not included and the reanalyses do
not simulate a severe storm. In his talk 'Improving Atmospheric Reconstructions For Historical Extreme
Events', Ed Hawkins shows the added value of such missing historical data for the severe

storm in 1903 (https://www.rmets.org/event/virtual-meeting-improving-atmospheric-reconstructions-historicalextreme- events)
For the severe storm 1906, there were data at the crucial regions already available and assimilated in the
reanalyse model. With the reconstruction, we can verify the pressure pattern with the reanalysis model data.
We addressed the points/concerns raised by the reviewer and revised these points in the chapters "1 Introduction" [page 3, lines 15-31 in the marked-up manuscript version] and "2.2. Reconstruction of sea level pressure and wind fields using the manual synoptic approach" [page 7, lines 23-27 in the marked- up
manuscript version] to make them clearer.
Hawkins et al., 2019, 'Hourly weather observations from the Scottish Highlands (1883–1904) rescued by
volunteer citizen scientists', Geoscience Data, 6, 160, doi: 10.1002/gdj3.79

I fully agree that data archaeology is extremely valuable; the reanalyses themselves are produced using historical observations.
The piece of text I was referring to (now on page 13, lines 15-22) was extended but the authors did not answer the question about similarity, and the reanalyses are still considered reliable because they look similar to the manual reconstructions. The authors seem to consider the manual reconstruction as a reference but I think that scientific results should have an objective basis, when available.

I suggest to say that, because the maps look similar, the main features of the respective wind fields are consistent.

We changed the text accordingly.  Clean text, on page 13, lines 21-22

Two inferences can be made: First, the similarity between the pressure charts suggests that the additional digitized data did not add value to the simulation of the wind fields compared to the used 20CRv3 reanalysis, because most of the new data are located in Central Europe and enough assimilation data in key regions were available for the reanalysis. Especially in the case of historical storm events, single missing pressure data can be crucial parameters for the course of isobars and thus incorrect wind speed can be calculated (Feuchter et al., 2013, Hawkins et al., 2019). Second, in regions with a high density
20 of assimilated data, ensemble variability in the reanalysis is substantially smaller compared to areas where no such data exists. The similarity between the pressure charts of the reconstruction and the reanalyses suggests that the main features of the respective wind are consistent.

3) 3.2 Ensemble simulations.

We split the figures to enhance the visibility. For the discussion it is important to make the most extreme water level visible. Therefore, we refrain from a synthetic assessment that will blur the individual results. [page 15- 16 in the marked-up manuscript version]

Ok, but the figures are still very busy, also because of the relatively large tidal signal. What about showing the de-tided curves and data?

We changed the colors of the ensemble members in grey. Only the maximum ensemble is highlighted. In addition, we included a table with maximum, median and standard deviation.

Clean text, page 14 - 16